# Explainability in Deep Learning Segmentation Models for Breast Cancer by Analogy with Texture Analysis

**Md Masum Billah**[1]                                    Md.Billah@abo.fi

**Pragati Manandhar**[1]                          Pragati.Manandhar@abo.fi

**Sarosh Krishan**[1]                              Sarosh.Krishan@abo.fi

**Alejandro Cedillo Gámez**[1]                   Alejandro.Cedillo@abo.fi

**Hergys Rexha**[1]                                   hergys.rexha@abo.fi

**Sebastien Lafond**[1]                                    slafond@abo.fi

**Kurt K. Benke**[2]                              kbenke@unimelb.edu.au

**Sepinoud Azimi**[3]                        s.azimirashti@tudelft.nl

**Janan Arslan**[4]                    janan.arslan@icm-institute.org

[1] *Faculty of Science and Engineering, Åbo Akademi University, Turku, Finland*

[2] *School of Engineering, University of Melbourne, Parkville, Victoria, Australia*

[3] *Technology, Policy and Management, Delft University of Technology, Delft, The Netherlands*

[4] *Sorbonne Université, Institut du Cerveau—Paris Brain Institute—ICM, CNRS, Inria, Inserm, AP-HP, Hôpital de la Pitié Salpêtrière, Paris, France*

**Editors:** Accepted for publication at MIDL 2024

## Abstract

Despite their predictive capabilities and rapid advancement, the black-box nature of Artificial Intelligence (AI) models, particularly in healthcare, has sparked debate regarding their trustworthiness and accountability. In response, the field of Explainable AI (XAI) has emerged, aiming to create transparent AI technologies. We present a novel approach to enhance AI interpretability by leveraging texture analysis, with a focus on cancer datasets. By focusing on specific texture features and their correlations with a prediction outcome extracted from medical images, our proposed methodology aims to elucidate the underlying mechanics of AI, improve AI trustworthiness, and facilitate human understanding. The code is available at https://github.com/xrai-lib/xai-texture.

**Keywords:** Artificial Intelligence, Cancer Diagnosis, Explainable AI, Texture Analysis, Medical Imaging

## 1. Introduction

Explainable Artificial Intelligence (XAI) was borne from the need to improve the transparency and interpretability of traditionally opaque AI models. Despite the advancements made by current XAI techniques (e.g., SHAP and Quantus), these methods provide only a partial understanding of the decision-making processes underlying AI-driven diagnoses. One promising avenue involves the utilization of Law's Texture Energy Measure (LTEM) as a feature extraction method, coupled with an Artificial Neural Network (ANN) to classify normal-abnormal and benign-malignant images (Setiawan et al., 2015). Previous research underscores that the classification accuracy strongly depends on the quality of the extracted texture features, indicating its potential as a tool for explainable classification

(Bouzar-Benlabiod et al., 2023). This study endeavors to leverage advanced texture analysis methods alongside AI segmentation models to elucidate how AI models arrive at their decisions, charting a course toward greater transparency and interpretability in medical AI technologies.

## 2. Methodology

**Dataset and Segmentation Models:** The open-source curated Breast Imaging Subset of the Digital Database for Screening Mammography (CBIS-DDSM) dataset was used (Lee et al., 2017). Patches of both the image and its equivalent mask containing the region of interest (ROI) were used in deep learning (DL) training instead of the full-sized images. The dataset was chosen for its detailed annotations and minimal class imbalance. The DL models selected for training were U-Net, for its precise localization due to a symmetric architecture (Ronneberger et al., 2015), DeepLabv3, for its atrous spatial pyramid pooling components which achieve a higher accuracy (Chen et al., 2017), and FCN for its deep feature extraction capabilities (He et al., 2016). The backbone of FCN and DeepLabv3 is ResNet101.

**Law's Texture Energy Measure [LTEM]:** The LTEM utilizes four primary image characteristics: **Level (L5)**, **Edge (E5)**, **Spot (S5)**, and **Ripple (R5)** represented as vectors, combined two at a time, to create 5x5 masks (Fig. 1 [A]). When applied to cancer images (Fig. 1 [B]), they extract specific texture features (Fig. 1 [C]).

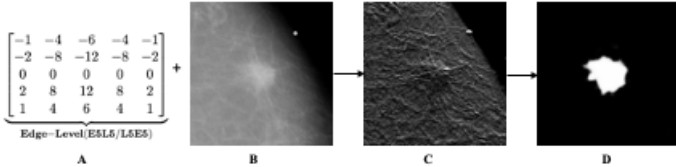

Figure 1: LTEM L5E5-based DeepLabv3 training.

The combination of the base four LTEMs led to the generation of 16 prospective textures. Some combinations were symmetrical, as they were mirrors of each other. Thus, there are a total of 9 unique texture masks (Setiawan et al., 2015). The LTEMs were applied to the cropped CBIS-DDSM dataset to create 9 datasets, each highlighting a specific texture energy measure. Ten copies of each model were trained for the 9 LTEM datasets, along with the original cropped dataset, giving us 30 models in total (i.e., [raw data + 9 LTEM data] * 3 DL models).

**Gray Level Co-occurrence Matrix [GLCM]:** GLCM - a set of texture-based statistical measures - was additionally assessed, including Angular Second Moment (ASM), Contrast, Correlation, Variance, Inverse Difference Moment (IDM), Sum Average, Sum Entropy, Entropy, Difference Entropy, Information Measure of Correlation 1 (IMC1), Information Measure of Correlation 2 (IMC2) and Autocorrelation (Rout et al., 2022). These measures were extracted for both raw input images and the final layer feature maps of the trained DL models.

| Rank | DeepLabv3 | | FCN | | U-Net | |
|------|-----------|--|-----|--|-------|--|
| | **IoU** | **CS** | **IoU** | **CS** | **IoU** | **CS** |
| | Original Model IoU: 87.02 | | Original Model IoU: 84.86 | | Original Model IoU: 62.65 | |
| 1 | L5E5: 82.64 | L5E5: 0.3037 | L5E5: 81.80 | L5R5: 0.3212 | L5E5: 60.38 | L5R5: 0.3019 |
| 2 | E5E5: 73.85 | E5E5: 0.2461 | E5E5: 74.60 | L5E5: 0.2946 | L5S5: 43.83 | E5E5: 0.2785 |
| 3 | L5S5: 73.43 | L5R5: 0.2345 | L5S5: 72.78 | R5S5: 0.2337 | S5S5: 43.42 | L5S5: 0.2593 |

Table 1: LTEM ranked by feature map comparison using Intersection over Union (IoU) and cosine similarities (CS)

| Rank | DeepLabv3 | FCN | U-Net |
|------|-----------|-----|-------|
| 1 | IMC1: 0.00075 | IMC1: 0.00060 | IMC1: 0.00260 |
| 2 | Autocorrelation: 0.02375 | Autocorrelation: 0.02586 | MCC: 0.19584 |
| 3 | ASM: 0.07456 | ASM: 0.15945 | Autocorrelation: 0.20374 |

Table 2: GLCM features ranked by significance

## 3. Results and Discussion

The DL models were first ranked based on Intersection over Union (IoU) results (Table 1). For LTEM-based models, those trained on the **L5E5** feature performed the best, followed by **E5E5** for DeepLabv3 and FCN, and **L5S5** for U-Net. Third place was shared by **L5S5** for DeepLabv3 and FCN, and **S5S5** for the U-Net. This indicates a clear pattern that the **Level** and **Edge** features of the cancer images contain the most useful information. Second, using cosine similarity (CS), we compared the feature maps generated by the LTEM models with the models trained on the original dataset (Table 1). In this analysis, the **L5E5, E5E5** and **L5R5** models generated feature maps more similar to the original model, reaffirming the significance of the **Level** and **Edge** features. The GLCM features of the raw images were compared to those derived from the feature maps, utilizing average absolute differences as the metric of comparison. Table 2 highlights the top three features for each DL model. There was a consistent pattern across all models, identifying IMC1 as the most critical GLCM feature. IMC1 measures the degree to which the joint entropy of pixel pairs in the image is reduced compared to the entropy of the individual pixels. Additionally, Autocorrelation, along with ASM and MCC also emerged as significant features influencing model behavior.

Both quantitatively and qualitatively (Fig 1 [D]), we can see texture-driven DL models (particularly **L5E5**) have comparable performance to DL models fed with raw data, reiterating the importance of these features in DL training. With respect to explainability, the results suggest that the DL algorithm is responding to texture periodicity (size of the repeating fundamental pattern) and edges, particularly horizontal edge structure embedded in the textures.

**Acknowledgments:** The work has been partially supported by the EMJMD master's program in Engineering of Data-Intensive Intelligent Software Systems (EDISS - European Union's Education, Audiovisual and Culture Executive Agency grant number 619819).

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
