# OpenReview forum: "Explainability in Deep Learning Segmentation Models for Breast Cancer by Analogy with Texture Analysis"
_MIDL.io/2024/Short_Papers — MIDL 2024 Short Papers_

### Official Review · Reviewer_bEjC · 2024-04-24

**Confidence:** 5
**Final Rating:** 4

**Review:**

This research proposes a novel approach to improve the interpretability of Artificial Neural Networks (ANNs) used for cancer diagnosis in medical imaging. The authors leverage texture analysis of medical images alongside ANNs, focusing on specific texture features extracted from images and how they correlate with the ANN's prediction. By analyzing these relationships, they aim to understand what aspects of the image the ANN is using to make its decision.
The authors use a publicly available breast cancer image dataset (CBIS-DDSM) and train three different deep learning segmentation models (U-Net, DeepLabv3, FCN) on the dataset. For texture analysis, they apply Law's Texture Energy Measure (LTEM) to the images, generating different texture feature maps, and extract Gray Level Co-occurrence Matrix (GLCM) features from both raw images and the final feature maps of the trained models.
This work offers a promising approach for explaining ANN decisions in medical imaging by using texture analysis. By understanding which image features are most important for the ANN, researchers can gain insights into the model's reasoning and improve trust in its predictions.

However, the research is in its initial stages, and further validation on larger and more diverse datasets is needed. The chosen texture analysis methods (LTEM, GLCM) might not capture all the complexities influencing ANN decisions, and it might be challenging to translate these texture-based explanations into easily understandable terms for medical professionals. Even if texture features show a correlation with the model's performance, it doesn't directly explain the internal workings of the ANN. It might only reveal which image characteristics are important but not how the ANN reasons about them.

---

### Decision · Program_Chairs · 2024-04-26

Accept